# An Interactive Consensus Model in Group Decision Making with Heterogeneous Hesitant Preference Relations

**Yongming Song**

School of Business Administration, Shandong Technology and Business University, Yantai 264005, China; songyongming@sdtbu.edu.cn

**Abstract:** This paper proposes an interactive consensus reaching model in the group decision making for heterogeneous hesitant preference relations (i.e., hesitant fuzzy preference relations, hesitant multiplicative preference relations, hesitant fuzzy linguistic preference relations). First, the consistencies of three hesitant preference relations are defined, respectively. Then, based on their definitions, three optimization models are constructed to obtain the weight vector of alternatives, based on which an interactive consensus adjustment algorithm is established based on the direct consensus framework. This framework adopts feedback mechanism to facilitate the information correction of decision makers. After several rounds of adjustment, the decision results with satisfactory consensus level are achieved. Finally, the practicability and effectiveness of the model are illustrated through a case study of mine accident emergency decision making.

**Keywords:** group decision making; heterogeneous hesitant preference relations; consensus reaching process; mathematical programming; emergency decision-making

**MSC:** 90B50; 91B06

## 1. Introduction

Decision making is a very important step in management activities [1,2]. All of these decisions are evaluation-based personal decision options, usually based on the preferences, experience, and other data about the decision maker [3,4]. With more and more complex decision-making environments, the decision-making activities of modern human beings involve a wide range of information and many influencing factors, making it impossible to achieve scientific decision-making with only a single person's ability. Scientific decision-making needs to concentrate the advantages and wisdom of group to make the best decision. Therefore, group decision-making (GDM) is valued in society and organizations. GDM is conducive to pooling the wisdom of experts in different fields and making full use of their members' different professional knowledge, experience and background to improve the comprehensiveness and science of decision-making [5]. A GDM problem can be defined as a process in which multiple decision makers (or experts) choose the most appropriate solution from two or more possible alternatives [6]. Many approaches have been proposed to work out GDM problems [7–12]. In these studies, the decision maker is allowed to use only one form of preference structure. However, in practical GDM problems, decision makers have different educational backgrounds, experiences, and cultures. Therefore, different decision makers may use different preference structures to represent preference information [13]. The commonly used preference structures have preference orderings, utility values, multiplicative preference relations (MPRs), fuzzy preference relations (FPRs), and linguistic preference relations (LPRs). Herrera-Viedma et al. [14] presented a consensus model with preference ordering, FPRs, MPRs, and utility function based on two consensus criteria. Fan et al. [15] established a linear goal programming model to deal with MPRs and FPRs. A consensus model was proposed where decision makers can use numerical, linguistic and interval-valued information to express their opinions [16]. Dong and Zhang [17]

proposed a direct consensus method with MPRs, FPRs, preference orderings, and utility functions. Zhang and Guo [18] constructed a GDM method with heterogeneous incomplete uncertain preference relations (i.e., uncertain MPRs, uncertain FPRs, uncertain LPRs and intuitionistic fuzzy preference relations). Tang et al. [19] proposed a new GDM method based on the consistency-consensus optimization model to deal with incomplete heterogeneous preference relations (i.e., FPRs, MPRs, additive LPRs, and multiplicative LPRs). Kou et al. [20] proposed a geometrical method to reach consensus with heterogeneous preference information.

In the actual decision-making process, when facing the increasingly complex decision-making environment, people tend to be in a state of hesitation when evaluating complex problems due to the lack of knowledge and experience, time pressure, and other factors. In order to resolve situations in which hesitation occurs, Torra [21] proposed the concept of a hesitant fuzzy set, which allows decision makers at the same time considering several possible values to express their opinion for target of evaluation. For example, someone is invited to evaluate the risk of an investment scheme. The higher the value of the evaluation, the higher the risk. A full score of 1 indicates the very high risk. If he/she hesitates in the three evaluation values, i.e., 0.6, 0.7, and 0.8, the evaluator's evaluation information can be expressed as a hesitant fuzzy element {0.6, 0.7, 0.8}. Based on this hesitant fuzzy set, Rodriguez et al. [22] proposed hesitant fuzzy linguistic terms set (HFLTS), which improved the richness of language expression. Furthermore, Xia et al. [23] proposed hesitant fuzzy preference relation (HFPR) and hesitant multiplicative preference relation (HMPR). Zhu et al. [24] proposed hesitant fuzzy linguistic preference relation (HFLPR). Based on which, a large number of studies have been conducted to establish the consistency conditions and group consensus with HFPRs, HMPRs, and HFLPRs. Zhang and Dai [25] presented a consistency improvement model with HFLPRs based on a new consistency concept. Rehman et al. [26] constructed an improved consensus-based procedure to handle GDM for HFPRs. Li et al. [27] focused on building consensus with HFPRs based on multiplicative consistency. Liu et al. [28] established a new efficient consistency–consensus framework for HFLPRs. As can be seen from the literature review above, the consistency and consensus are two prominent parts in GDM problems. Lack of consistency will lead to unreasonable results and negatively affect the outcome of decision making [29]. Therefore, consensus building process has become an important research content in GDM, and a large number of research results have emerged [30–33].

The most relevant research in this study is GDM with heterogeneous hesitant preference relations (HHPRs). Zhang and Guo [34] focused on the fusion of incomplete HHPRs (i.e., HFPRs and HMPRs) under group decision making settings, without considering consensus-building process. He and Xu [35] proposed a mathematic programming model to process HFPRs, HMPRs, and the hesitant preference orderings. The HHPRs provide effective tools to represent preference information and improve the quality of decision. However, the existing methods cannot deal with the HHPRs including linguistic type-based hesitant preference relations, which limits the scope of their application.

Based on the above analysis, the objective of this study is to propose a consensus reaching process in GDM based on HHPRs (including HFPRs, HMPRs, and HFLPRs). Firstly, according to the consistency definitions of three hesitant preference relations, three optimization models are established to obtain the priority weight of the alternatives, respectively. Based on this, IOWA operator assembly is used to obtain the priority weight of the group. Then, an interactive group consensus reaching process is established. Finally, the feasibility and effectiveness of the proposed method are illustrated by an emergency decision-making problem.

There are three key contributions of this paper. First, the consensus model constructed in this paper can deal with HHPRs including not only numerical value-based hesitant preference relations (HFPRs and HMPRs), but also linguistic-based hesitant preference relations (HFLPRs), which further expands the application of the HHPRs in the GDM. Second, three optimization models based on consistencies of HHPRs are established to deal

with HHPRs and obtain the priority weights of alternatives, based on which an interactive consensus adjustment algorithm is established according to the direct consensus framework. Third, we construct the weights of decision makers according to the consistency bias and the experts' weights are dynamically adjusted based on their information quality in the process of consensus adjustment.

The rest of this article is organized as follows. Section 2 gives the preliminaries about this study. In Section 3, the framework of this study's method is established about consensus reaching process with HHPRs. In Section 4, our method is used to solve a rescue plan selection problem. In Section 5, the comparative analysis, managerial implications, and conclusions of this paper are given. Finally, Section 6 summarizes the conclusion of this paper and the future research direction.

## 2. Preliminaries

### 2.1. Two Tuple Linguistic Expressive Model

Let $S = \{s_0, s_1, \cdots, s_g\}$ be a linguistic term set with odd granularity and $g + 1$ is the granularity of the linguistic $S$. Herrera and Martinez [36] presented the 2-tuple linguistic expressive model $(s_i, \alpha_i)$.

**Definition 1 ([36]).** *Suppose $\beta \in [0, g]$ is the result of a symbolic aggregation operation in a linguistic term set $S = \{s_0, s_1, \cdots, s_g\}$. Then, the equivalent information to $\beta$ in the 2-tuple is obtained by using the following function:*

$$\Delta : [0, g] \rightarrow S \times [-0.5, 0.5)$$
$$\Delta(\beta) = \begin{cases} s_i & i = round(\beta) \\ \alpha = \beta - i & \alpha \in [-0.5, 0.5) \end{cases} \tag{1}$$

**Definition 2 ([36]).** *Suppose $S = \{s_0, s_1, \cdots, s_g\}$ is a linguistic term set and $(s_i, \alpha)$ is a 2-tuple, then there exists a function $\Delta^{-1}$ which transforms a 2-tuple into its equivalent numerical value $\beta \in [0, g]$. The function $\Delta^{-1}$ is defined as follows:*

$$\Delta^{-1} : S \times [-0.5, 0.5) \rightarrow [0, g]$$
$$\Delta^{-1}(s_i, \alpha) = i + \alpha \tag{2}$$

In addition, a linguistic label would be changed into the representation of a 2-tuple linguistic according to Definitions 1 and 2 by adding a zero as a symbolic translation, i.e., $\Delta(s_i) = (s_i, 0)$.

### 2.2. Heterogeneous Hesitant Preference Relations and Their Consistencies

In the following, we review the definitions of HMPR, HFPR, and HFLPR.

**Definition 3 ([23]).** *Given a set of alternatives $X = \{x_1, x_1, \cdots, x_n\}$, the HMPR of alternatives is expressed as $H = (h_{ij})_{n \times n} \subset X \times X$, where $h_{ij} = \left\{ r_{ij}^{\sigma(l)} | l = 1, 2, \cdots, \#h_{ij} \right\}$ ($\#h_{ij}$ indicates number of elements in $h_{ij}$) and also represents a hesitant multiplicative element, indicating the degree of preference over alternative $x_i$ to $x_j$. In addition, $h_{ij}$ meets the following conditions.*

$$r_{ij}^{\sigma(l)} \cdot r_{ji}^{\sigma(\#h_{ij}-l+1)} = 1, \ r_{ii} = \{1\}, \ \#h_{ij} = \#h_{ji}, \ i, j = 1, 2, \cdots, n \tag{3}$$

*where $r_{ij}^{\sigma(l)}$ indicates the lth element in $h_{ij}, r_{ij}^{\sigma(l)} \in \left[ \frac{1}{9}, 9 \right]$.*

**Definition 4 ([23]).** *Given a set of alternatives $X = \{x_1, x_1, \cdots, x_n\}$, the HFPR of alternatives is expressed as $H = (h_{ij})_{n \times n} \subset X \times X$, where $h_{ij} = \left\{ r_{ij}^{\sigma(l)} | l = 1, 2, \cdots, \#h_{ij} \right\}$ ($\#h_{ij}$ indicates*

*number of elements in $h_{ij}$) and also represents a hesitant fuzzy element, indicating the degree of preference over alternative $x_i$ to $x_j$; $h_{ij}$ meets the following conditions.*

$$r_{ij}^{\sigma(l)} + r_{iji}^{\sigma(\#h_{ij}-l+1)} = 1, \; r_{ii} = \{0.5\}, \; \#h_{ij} = \#h_{ji}, \; i,j = 1,2,\cdots,n \tag{4}$$

*where $r_{ij}^{\sigma(l)}$ indicates the lth element in $h_{ij}$, $r_{ij}^{\sigma(l)} \in [0,1]$.*

**Definition 5 ([24]).** *Given a set of alternatives $X = \{x_1, x_1, \cdots, x_n\}$, the HFLPR of alternatives is expressed as $H = (h_{ij})_{n \times n} \subset X \times X$, satisfying the following conditions for $i,j = 1,2,\cdots,n$, $h_{ij}(i < j)$.*

$$\Delta^{-1}(\Delta\left(h_{ij}^{\sigma(l)}\right) + \Delta\left(h_{ji}^{\sigma(l)}\right)) = g, \; h_{ii} = \{s_{g/2}\}, \; \#h_{ij} = \#h_{ji} \tag{5}$$

$$h_{ij}^{\sigma(l)} < h_{ij}^{\sigma(l+1)}, h_{ji}^{\sigma(l+1)} < h_{ji}^{\sigma(l)} \tag{6}$$

*where $h_{ij} = \left\{ h_{ij}^l | l = 1,2,\cdots,\#h_{ij} \right\}$ ($\#h_{ij}$ indicates number of elements in $h_{ij}$) as a hesitant linguistic element, showing the set of all possible preferred language values $x_i$ as superior to $x_j$. $h_{ij}^{\sigma(l)}$ as indicated the lth element in $h_{ij}$, $h_{ij}^{\sigma(l)} \in S = \{s_0, s_1, \cdots, s_g\}$.*

Moreover, the consistency condition of HFPR is given as follows:

**Definition 6 ([37]).** *Suppose $H = (h_{ij})_{n \times n}$ be a HFPR, if $H$ meets the following conditions, then $H$ is an additive consistent HFPR.*

$$0.5(w_i - w_j) + 0.5 = h_{ij}^{(1)} \text{ or } h_{ij}^{(2)} or \cdots or \; h_{ij}^{(\#h_{ij})} \tag{7}$$

*where $w = (w_1, w_2, \cdots, w_n)^T$ indicates priority weight vector of $H$, $\sum_{i=1}^n w_i = 1, w_i \geq 0$, $i,j = 1,2,\cdots,n$.*

For a multiplicative preference relation $A^* = \left(a_{ij}^*\right)_{n \times n}$, if the following equation is true, then $A^*$ is perfectly consistent [38].

$$a_{ij}^* = \frac{w_i}{w_j} \tag{8}$$

Based on the Equation (8), the consistent definition of HMPR is given below.

**Definition 7.** *Suppose $A = (a_{ij})_{n \times n}$ be an HMPR, if $A$ meets the following conditions, then $A$ is a multiplicative consistent HMPR.*

$$\frac{w_i}{w_j} = a_{ij}^{(1)} \text{ or } a_{ij}^{(2)} or \cdots or \; a_{ij}^{(\#a_{ij})} \tag{9}$$

*where $w = (w_1, w_2, \cdots, w_n)^T$ indicates priority weight vector of $A$, $\sum_{i=1}^n w_i = 1, w_i \geq 0$, $i,j = 1,2,\cdots,n$.*

For a linguistic preference relation $T^* = \left(t_{ij}^*\right)_{n \times n}$, if the following equation is true, then $T^*$ is additive consistent [39,40]

$$\Delta^{-1}\left(t_{ij}^*\right) = \frac{gn}{2}(w_i - w_j) + \frac{g}{2} \tag{10}$$

Based on the consistent condition of linguistic preference relation (i.e., Equation (10)), the consistent definition of HFLPR is given below.

**Definition 8.** *Suppose* $T = (t_{ij})_{n \times n}$ *be a HFLPR, if T meets the following conditions, then T is an additive consistent HFLPR.*

$$\frac{gn}{2}(w_i - w_j) + \frac{g}{2} = \Delta^{-1}(t_{ij}^{(1)}) or \ \Delta^{-1}(t_{ij}^{(2)}) or \cdots or \ \Delta^{-1}(t_{ij}^{(\#t_{ij})}) \tag{11}$$

*where* $w = (w_1, w_2, \cdots, w_n)^T$ *indicates priority weight vector of T,* $\sum_{i=1}^{n} w_i = 1, w_i \geq 0,$ $i, j = 1, 2, \cdots, n.$

## 3. The Framework of Consensus Reaching Process with HHPRs

It is assumed decision-makers $e_k(k = 1, 2, \cdots, m_3)$ may use HFPR, HMPR, and HFLPR to express their preference information for the alternatives respectively, indicated by $H_i(i = 1, 2, \cdots, m_1)$, $A_i(i = m_1 + 1, m_1 + 2, \cdots, m_2)$, and $T_i(i = m_2 + 1, m_2 + 2, \cdots, m_3)$. In addition, $w^c = (w_1^c, w_2^c, \cdots, w_n^c)^T$ indicates weight vector of group for alternatives $X = (x_1, x_2, \cdots, x_n)$. There are two processes to go through before the final solution is reached: the selection phase and the consensus reaching process.

### 3.1. The Selection Phase

In this stage, the individual priority weight vector is firstly obtained, and then the group priority weight vector is obtained by IOWA operator assembly.

3.1.1. Obtaining the Individual Priority Weight Vector

According to the consistencies of HHPRs, three optimization models with the minimum deviation are established to obtain the individual priority weight vector under HHPRs. The following three cases are discussed.

(1)    $e_k \in E^H$

For a HFPR $H = (h_{ij})_{n \times n}$, let $\delta\left(h_{ij}^{\sigma(l)}\right) = h_{ij}^{(1)}$ or $h_{ij}^{(1)}$ or $\cdots$ or $h_{ij}^{(\#h_{ij})}$, thus the Equation (7) can be written to:

$$0.5(w_i - w_j) + 0.5 = \delta\left(h_{ij}^{\sigma(l)}\right) \Leftrightarrow 0.5(w_i - w_j) + 0.5 - \delta\left(h_{ij}^{\sigma(l)}\right) = 0 \tag{12}$$

If a HFPR is not additive consistent, then there is no weight vector satisfying Equation (12). However, it is difficult for decision-makers to give HFPR with perfect consistency. In this case, "soft consistency" [41] was proposed to represent approximate consistency. In order to get the most appropriate weight vector, we minimize the total deviation $\left|0.5(w_i - w_j) + 0.5 - \delta\left(h_{ij}^{\sigma(l)}\right)\right|$. Considering that the weight vector is non-negative and the sum is 1, the following mathematical programming is established to find the weight vector:

$$\begin{aligned} \min \varepsilon_{ij} &= \left|0.5\left(w_i^k - w_j^k\right) + 0.5 - \delta\left(h_{ij}^{\sigma(l)}\right)\right| \\ s.t. \sum_{t=1}^{n} w_t^k &= 1, \ 0 \leq w_i^k, w_j^k \leq 1, i, j = 1, 2, \cdots, n \end{aligned} \tag{13}$$

And because this is true

$$\begin{aligned} &\left|0.5\left(w_j^k - w_i^k\right) + 0.5 - \delta\left(h_{ji}^{\sigma(l)}\right)\right| \\ &= \left|0.5\left(w_j^k - w_i^k\right) + 0.5 - \left(1 - \delta\left(h_{ij}^{\sigma(l)}\right)\right)\right| \\ &= \left|0.5\left(w_j^k - w_i^k\right) - 0.5 + \delta\left(h_{ij}^{\sigma(l)}\right)\right| \\ &= \left|0.5\left(w_i^k - w_j^k\right) + 0.5 - \delta\left(h_{ij}^{\sigma(l)}\right)\right| \end{aligned}$$

Therefore, model (13) can be simplified to the following mathematical program (14)

$$\min \varepsilon_{ij} = \left| 0.5\left( w_i^k - w_j^k \right) + 0.5 - \delta\left( h_{ij}^{\sigma(l)} \right) \right|$$
$$s.t. \sum_{t=1}^{n} w_t^k = 1, \ 0 \le w_i^k, w_j^k \le 1, i, j = 1, 2, \cdots, n, j > i \tag{14}$$

Further, as $\delta\left( h_{ij}^{\sigma(l)} \right) = h_{ij}^{(1)} \ or \ h_{ij}^{(1)} or \cdots or \ h_{ij}^{(\#h_{ij})}$, the model (14) can be converted into the following mathematical programming

$$\min F = \sum_{i=1}^{n-1} \sum_{j=i+1}^{n} s_{ij} d_{ij,k}^+ + t_{ij} d_{ij,k}^-$$

$$s.t. \begin{cases} 0.5\left( w_i^k - w_j^k \right) + 0.5 - \sum_{l=1}^{\#h_{ij,k}} z_{ij,k}^{\sigma(l)} h_{ij,k}^{\sigma(l)} - d_{ij,k}^+ + d_{ij,k}^- = 0 \\ \sum_{i=1}^{n} w_i^k = 1, \ w_i^k \ge 0, i = 1, 2, \cdots, n \\ \sum_{l=1}^{\#h_{ij,k}} z_{ij,k}^{\sigma(l)} = 1, \ i, j = 1, 2, \cdots, n, \ j > i, k = 1, 2, \cdots, m_1 \\ z_{ij,k}^{\sigma(l)} = 0 \ or \ 1, i, j = 1, 2, \cdots, n, \ j > i, k = 1, 2, \cdots, m_1 \\ d_{ij,k}^+, d_{ij,k}^- \ge 0, \ i, j = 1, 2, \cdots, n, \ j > i, k = 1, 2, \cdots, m_1 \end{cases} \tag{15}$$

where $s_{ij}$ and $t_{ij}$ represent the importance of positive deviation $d_{ij,k}^+$ and minus deviation $d_{ij,k}^-$, respectively.

Without loss of generality, all target functions are equal, i.e., $s_{ij} = t_{ij} = 1 (i, j \in N)$. Thus, the model (15) is translated to model (16).

$$\min F = \sum_{i=1}^{n-1} \sum_{j=i+1}^{n} d_{ij,k}^+ + d_{ij,k}^-$$

$$s.t. \begin{cases} 0.5\left( w_i^k - w_j^k \right) + 0.5 - \sum_{l=1}^{\#h_{ij,k}} z_{ij,k}^{\sigma(l)} h_{ij,k}^{\sigma(l)} - d_{ij,k}^+ + d_{ij,k}^- = 0 \\ \sum_{i=1}^{n} w_i^k = 1, \ w_i^k \ge 0, i = 1, 2, \cdots, n, k = 1, 2, \cdots, m_1 \\ \sum_{l=1}^{\#h_{ij,k}} z_{ij,k}^{\sigma(l)} = 1, \ i, j = 1, 2, \cdots, n, \ j > i, k = 1, 2, \cdots, m_1 \\ z_{ij,k}^{\sigma(l)} = 0 \ or \ 1, i, j = 1, 2, \cdots, n, \ j > i, k = 1, 2, \cdots, m_1 \\ d_{ij,k}^+, d_{ij,k}^- \ge 0, \ i, j = 1, 2, \cdots, n, \ j > i, k = 1, 2, \cdots, m_1 \end{cases} \tag{16}$$

Based on positive and negative deviation $d_{ij,k}^+, d_{ij,k}^-$, we define the consistency index $CI(e_k)$ of decision makers $e_k$ as

$$CI(e_k) = 1 - \frac{2\sum_{i=1}^{n-1} \sum_{j=2,j>i}^{n} \left( d_{ij}^+ + d_{ij}^- \right)}{n(n-1)} \tag{17}$$

(2)　For $e_k \in E^A$, let $\delta\left(a_{ij}^{\sigma(l)}\right) = a_{ij}^{(1)}$ or $a_{ij}^{(2)}$ or $\cdots$ or $a_{ij}^{(\#a_{ij})}$, then $\frac{w_i}{w_j} = a_{ij}^{(1)}$ or $a_{ij}^{(2)}$ or $\cdots$ or $a_{ij}^{(\#a_{ij})} \Leftrightarrow \frac{w_i}{w_j} = \delta\left(a_{ij}^{\sigma(l)}\right) \Leftrightarrow w_i - w_j \cdot \delta\left(a_{ij}^{\sigma(l)}\right) = 0$, in order to minimize the total deviation $\left| w_i - w_j \cdot \delta\left(a_{ij}^{\sigma(l)}\right) \right|$, we obtain the following optimization model

$$\min F = \sum_{i=1}^{n-1} \sum_{j=i+1}^{n} d_{ij,k}^+ + d_{ij,k}^-$$

$$s.t. \begin{cases} w_i^k - \left( \sum_{l=1}^{\#p_{ij,k}} z_{ij,k}^{\sigma(l)} a_{ij,k}^{\sigma(l)} \right) w_j^k - d_{ij,k}^+ + d_{ij,k}^- = 0 \\ \sum_{i=1}^{n} w_i^k = 1,\ w_i^k \geq 0, i = 1,2,\cdots,n, k = m_1+1, m_1+2,\cdots,m_2 \\ \sum_{l=1}^{\#p_{ij,k}} z_{ij,k}^{\sigma(l)} = 1,\ i,j = 1,2,\cdots,n,\ j > i, k = m_1+1, m_1+2,\cdots,m_2 \\ z_{ij,k}^{\sigma(l)} = 0 \text{ or } 1,\ i,j = 1,2,\cdots,n,\ j > i, k = m_1+1, m_1+2,\cdots,m_2 \\ d_{ij,k}^+, d_{ij,k}^- \geq 0,\ i,j = 1,2,\cdots,n,\ j > i, k = m_1+1, m_1+2,\cdots,m_2 \end{cases} \quad (18)$$

Based on positive and negative bias $d_{ij,k}^+, d_{ij,k}^-$, the consistency index $CI(e_k)$ of decision makers $e_k$ as

$$CI(e_k) = 1 - \frac{2\sum_{i=1}^{n-1}\sum_{j=2,j>i}^{n}\left(d_{ij}^+ + d_{ij}^-\right)}{n(n-1)} \quad (19)$$

(3)　For $e_k \in E^T$, let $\delta\left(\Delta^{-1}(t_{ij}^{\sigma(l)})\right) = \Delta^{-1}(t_{ij}^{(1)})$ or $\Delta^{-1}(t_{ij}^{(2)})$ or $\cdots$ or $\Delta^{-1}(t_{ij}^{(\#t_{ij})})$, then $\frac{gn}{2}(w_i - w_j) + \frac{g}{2} = \delta\left(\Delta^{-1}(t_{ij}^{\sigma(l)})\right) \Leftrightarrow \frac{gn}{2}(w_i - w_j) + \frac{g}{2} - \delta\left(\Delta^{-1}(t_{ij}^{\sigma(l)})\right) = 0$, in order to minimize total deviation $\left| \frac{gn}{2}(w_i - w_j) + \frac{g}{2} - \delta\left(\Delta^{-1}(t_{ij}^{\sigma(l)})\right) \right|$, the following mathematical programming is established:

$$\min F = \sum_{i=1}^{n-1} \sum_{j=i+1}^{n} d_{ij,k}^+ + d_{ij,k}^-$$

$$s.t. \begin{cases} \frac{gn}{2}\left(w_i^k - w_j^k\right) + \frac{g}{2} - \left( \sum_{l=1}^{\#t_{ij,k}} z_{ij,k}^{\sigma(l)} \Delta^{-1}\left(t_{ij,k}^{\sigma(l)}\right) \right) - d_{ij,k}^+ + d_{ij,k}^- = 0 \\ \sum_{i=1}^{n} w_i^k = 1,\ w_i^k \geq 0, i = 1,2,\cdots,n, k = m_2+1, m_2+2,\cdots,m_3 \\ \sum_{l=1}^{\#t_{ij,k}} z_{ij,k}^{\sigma(l)} = 1,\ i,j = 1,2,\cdots,n,\ j > i, k = m_2+1, m_2+2,\cdots,m_3 \\ z_{ij,k}^{\sigma(l)} = 0 \text{ or } 1,\ i,j = 1,2,\cdots,n,\ j > i, k = m_2+1, m_2+2,\cdots,m_3 \\ d_{ij,k}^+, d_{ij,k}^- \geq 0,\ i,j = 1,2,\cdots,n,\ j > i, k = m_2+1, m_2+2,\cdots,m_3 \end{cases} \quad (20)$$

Based on positive and negative bias $d_{ij,k}^+, d_{ij,k}^-$, the consistency index $CI(e_k)$ of decision makers $e_k$ as

$$CI(e_k) = 1 - \frac{2\sum_{i=1}^{n-1}\sum_{j=2,j>i}^{n}\left(d_{ij}^+ + d_{ij}^-\right)}{gn(n-1)} \quad (21)$$

3.1.2. Obtaining the Group Priority Weight Vector

According to the IOWA operator with consistency index [42], group priority weight vector $w^{(c)} = \left( w_1^{(c)}, w_2^{(c)}, \cdots, w_n^{(c)} \right)^T$ is obtained by aggregating individual priority weight vector $w^{(k)} = \left( w_1^{(k)}, w_2^{(k)}, \cdots, w_n^{(k)} \right)^T (k = 1, 2, \cdots, m)$.

$$
\begin{aligned}
w_i^{(c)} &= IOWA_{Q^c} \left( w_i^{(1)}, w_i^{(2)}, \cdots, w_i^{(m)} \right) \\
&= \Phi_W \left( \left\langle cl^1, w_i^{(1)} \right\rangle, \left\langle cl^2, w_i^{(2)} \right\rangle, \cdots, \left\langle cl^m, w_i^{(m)} \right\rangle \right) \\
&= \sum_{\tau=1}^{m} \lambda_\tau w_i^{(\tau)}
\end{aligned}
\tag{22}
$$

$$
cl^{\sigma(\tau-1)} \geq cl^{\sigma(\tau)}, \lambda_\tau = Q \left( \frac{\sum_{k=1}^{\tau} cl^{\sigma(\tau)}}{T} \right) - Q \left( \frac{\sum_{k=1}^{\tau-1} cl^{\sigma(\tau)}}{T} \right)
\tag{23}
$$

where $T = \sum_{k=1}^{m} cl^{\sigma(k)}$ and $cl^{\sigma(k)}$ is the *k*th larger value in $\left\{ cl^1, cl^2, \cdots, cl^m \right\}$.

*3.2. Consensus Reaching Process*

3.2.1. Group Consensus Degree

Generally speaking, the consensus degree in heterogeneous GDM is to measure the distance between the individual priority weight vector and the group priority weight vector. Based on this, the definition of group consensus degree (GCD) is given below.

**Definition 9.** *Suppose $w_k$ and $w_c$ be individual priority weight vector and group priority weight vector, respectively. In this instance, the GCD of decision maker $e_k$ is defined as*

$$
GCD(e_k) = 1 - \sqrt{\frac{1}{n} \sum_{i=1}^{n} (w_{ij,k} - w_{i,c})^2}
\tag{24}
$$

Thus, the GCD of All Decision Makers

$$
GCD\{e_1, e_2, \cdots, e_m\} = \frac{1}{m} \sum_{k=1}^{m} GCD(e_k)
\tag{25}
$$

if $GCD\{e_1, e_2, \cdots, e_m\} = 1$, then all the decision makers are in agreement with the collective; otherwise, the larger for $GCD\{e_1, e_2, \cdots, e_m\}$, the higher the degree of group consensus.

3.2.2. Feedback Adjustment

The goal of feedback adjustment is to provide adjustment suggestions with the decision-maker to modify his/her preference and improve the level of group consensus. An interactive mechanism provides decision-makers with the direction in which they need to modify preferences. The following three interactive feedback adjustment mechanisms are constructed based on different preference structures:

(1)　for $e_k \in E^H (k = 1, 2, \cdots, m_1)$,

$$
\begin{cases}
\overline{h_{ij,k}^{\sigma(l)}} \in \left[ \min\left\{ h_{ij,k}^{\sigma(l)}, h_{ij,c} \right\}, \max\left\{ h_{ij,k}^{\sigma(l)}, h_{ij,c} \right\} \right], & i < j \\
\overline{h_{ij,k}^{\sigma(l)}} = 1, & i = j \\
\overline{h_{ij,k}^{\sigma(l)}} = 1 - \overline{h_{ji,k}^{\sigma(l)}} & i > j
\end{cases}
\tag{26}
$$

where $h_{ij,c} = 0.5 \left( w_i^c - w_j^c \right) + 0.5, i, j = 1, 2, \cdots, n$.

(2)  for $e_k \in E^A (k = m_1 + 1, m_1 + 2, \cdots, m_2)$,

$$\begin{cases} \overline{a_{ij,k}^{\sigma(l)}} \in \left[ \min\left\{ a_{ij,k}^{\sigma(l)}, a_{ij,c} \right\}, \max\left\{ a_{ij,k}^{\sigma(l)}, a_{ij,c} \right\} \right], & i < j \\ \overline{a_{ij,k}^{\sigma(l)}} = 1, & i = j \\ \overline{a_{ij,k}^{\sigma(l)}} = \frac{1}{\overline{a_{ji,k}^{\sigma(l)}}} & i > j \end{cases} \tag{27}$$

where $a_{ij,c} = \frac{w_i^c}{w_j^c}, i, j = 1, 2, \cdots, n$.

(3)  for $e_k \in E^T (k = m_2 + 1, m_2 + 2, \cdots, m_3)$,

$$\begin{cases} \overline{t_{ij,k}^{\sigma(l)}} \in \left[ \min\left\{ t_{ij,k}^{\sigma(l)}, t_{ij,c} \right\}, \max\left\{ t_{ij,k}^{\sigma(l)}, t_{ij,c} \right\} \right], & i < j \\ \overline{t_{ij,k}^{\sigma(l)}} = 1, & i = j \\ \overline{t_{ij,k}^{\sigma(l)}} = 1 - \overline{t_{ji,k}^{\sigma(l)}} & i > j \end{cases} \tag{28}$$

where $t_{ij,c} = \frac{gn}{2}\left( w_i^c - w_j^c \right) + \frac{g}{2}, i, j = 1, 2, \cdots, n$.

Based on the above analysis, consensus reaching Algorithm 1 is established as follows.

---

**Algorithm 1:** A consensus reaching process in GDM with HHPRs

---

**Input:** Initial HHPRs and preset threshold of GCD ($\overline{GCD}$).

**Output:** The adjusted HHPRs and final group priority weight vector.

**Step 1:** Set $z = 0$, $H_k^{(z)} = \left( h_{ij,k}^{(z)} \right)_{n \times n} (k = 1, 2, \cdots, m_1)$,
$A_k^{(z)} = \left( a_{ij,k}^{(z)} \right)_{n \times n} (k = m_1 + 1, m_1 + 2, \cdots, m_2)$, $T_k^{(z)} = \left( t_{ij,k}^{(z)} \right)_{n \times n} (k = m_2 + 1, m_2 + 2, \cdots, m_3)$.

**Step 2:** The individual priority weight vector $w_k^{(z)} = \left( w_{1,k}^{(z)}, w_{2,k}^{(z)}, \cdots, w_{n,k}^{(z)} \right) (k = 1, 2, \cdots, m)$ and consistent level $cl_k^{(z)} (k = 1, 2, \cdots, m)$ are obtained according to the Equations (16)–(21).

**Step 3:** According to Equation (22), group priority weight vector $w_c^{(z)} = \left( w_{1,c}^{(z)}, w_{2,c}^{(z)}, \cdots, w_{n,c}^{(z)} \right)$ is obtained by IOWA operator.

**Step 4:** Obtaining $GCD\{e_1, e_2, \cdots, e_m\}$ according to Equation (25), if
$GCD\{e_1, e_2, \cdots, e_m\} \geq \overline{GCD}$, go directly to Step 6; Otherwise, proceed to the next step.

**Step 5:** The decision makers adjust their preferences according to Equations (26)–(28), respectively. Then, set $z = z + 1$, go back to step 2.

**Step 6:** Suppose $\overline{H^{(k)}} = H_z^{(k)} (k = 1, 2, \cdots, m_1)$, $\overline{A^{(k)}} = A_z^{(k)} (k = m_1 + 1, m_1 + 2, \cdots, m_2)$,
$\overline{T^{(k)}} = T_z^{(k)} (k = m_2 + 1, m_2 + 2, \cdots, m_3)$, and $w^* = w_c^{(z)}$.

---

Moreover, Figure 1 shows the structural framework of GDM model with HHPRs.

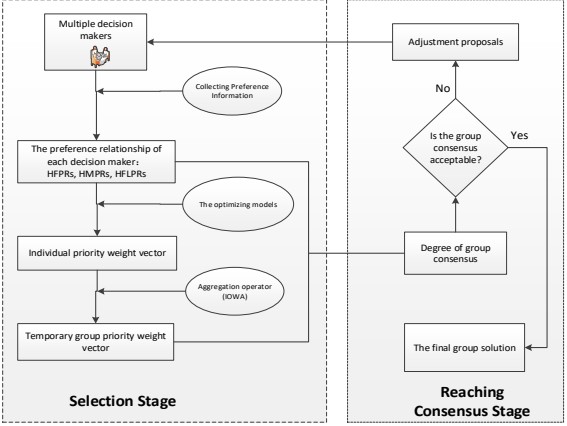

**Figure 1.** The framework of GDM with HHPRs.

### 4. Example Application: Emergency Plan Selection

In this section, we apply the proposed model to a real-world mining disaster rescue scheme selection. A collapse accident occurred in Pingyi coal mine in the city of Linyi in Shandong Province of China on 25 December 2015. A total of four miners escaped from the well and twenty-five were trapped underground. The municipal government quickly called a meeting with three experts $e_k (k = 1, 2, 3)$ ($e_1$: fire soldier, $e_2$: mine representative, and $e_3$: geological expert) to select the best rescue scheme from four alternatives $x_i (i = 1, 2, 3, 4)$ as follows:

(a)    Mining rescue channel and escape passage in the style of roadway drivage underground ($x_1$);
(b)    Dispatching large mechanical equipment and deep-hole drilling machines above mine ($x_2$);
(c)    Repair the wellbore and take mine cars down into the mine ($x_3$);
(d)    Using partial blasting and arranging mining machines ($x_4$).

Because the decision makers need to make decisions in a short amount of time to ensure the safety of miners' lives, in this case, the decision makers are experiencing hesitation at present. Based on this decision, using HHPRs are more consistent with the emergency decision scenario. Furthermore, three experts $e_k (k = 1, 2, 3)$ come from different backgrounds, therefore allowing each expert to use their own familiar characterization method to evaluate the four rescue plans. It is assumed that expert 1 adopts HFPR, expert 2 adopts HMPR, and expert 3 uses HFLPR. By comparing each pair of rescue schemes, the three experts construct the following preferences for the four rescue schemes as Tables 1–3.

**Table 1.** The HFPR given by expert $e_1$.

|       | $x_1$      | $x_2$      | $x_3$  | $x_4$      |
|-------|------------|------------|--------|------------|
| $x_1$ | {0.5}      | {0.3, 0.4} | {0.4}  | {0.6}      |
| $x_2$ | {0.7, 0.6} | {0.5}      | {0.5}  | {0.5, 0.6} |
| $x_3$ | {0.6}      | {0.5}      | {0.5}  | {0.6}}     |
| $x_4$ | {0.3}      | {0.5, 0.4} | {0.3}  | {0.5}      |

**Table 2.** The HMPR given by expert $e_2$.

|       | $x_1$       | $x_2$ | $x_3$      | $x_4$  |
|-------|-------------|-------|------------|--------|
| $x_1$ | {1}         | {1/3} | {3}        | {3, 5} |
| $x_2$ | {3}         | {1}   | {3}        | {5}    |
| $x_3$ | {1/3}       | {0.5} | {1}        | {3, 4}}|
| $x_4$ | {1/3, 1/5}  | {1/5} | {1/3, 1/4} | {1}    |

**Table 3.** The HFLPR given by expert $e_3$.

|       | $x_1$        | $x_2$        | $x_3$        | $x_4$        |
|-------|--------------|--------------|--------------|--------------|
| $x_1$ | {$s_3$}      | {$s_3$}      | {$s_3, s_4$} | {$s_4$}      |
| $x_2$ | {$s_5$}      | {$s_3$}      | {$s_5$}      | {$s_5, s_6$} |
| $x_3$ | {$s_5, s_4$} | {$s_3$}      | {$s_3$}      | {$s_5$}}     |
| $x_4$ | {$s_4$}      | {$s_3, s_2$} | {$s_3$}      | {$s_3$}      |

In what follows, the detailed realization of the model is illustrated.

(1)    Obtaining the priority weight vector of individual

According to the Equations (16)–(21), we establish three goal programs and get priority weight vectors and the consistent levels of the three experts $e_k (k = 1, 2, 3)$ as follows

$$w_1 = (0.15, 0.35, 0.35, 0.15)^T, cl_1 = 0.9833$$
$$w_2 = (0.3947, 0.3947, 0.1316, 0.0789)^T, cl_2 = 0.9386$$
$$w_3 = (0.2643, 0.3191, 0.2643, 0.1524)^T, \ cl_3 = 0.9167$$

(2) Obtaining the group priority weight vector

In this paper, IOWA operator is implemented by $Q$ function $Q(x) = x^{0.9}$. As $cl_1 = 0.9833$, $cl_2 = 0.9386$ and $cl_3 = 0.9167$, we obtain $\sigma(1) = 1, \sigma(2) = 2, \sigma(3) = 3$ according to Equation (23). Then, the weights of the three decision makers are obtained based on $Q(x) = x^{0.9}$ as follows:

$$\lambda_1 = 0.3851, \ \lambda_2 = 0.3189, \ \lambda_3 = 0.296$$

Based on the weights of the three decision makers, IOWA operator is used to aggregate and obtain the priority weight vector of the group $w_c^{(0)} = (0.2619, 0.3551, 0.255, 0.128)^T$.

(2) Consensus reaching process

The realization of consensus process requires two steps: consensus measurement and feedback adjustment. According to Equation (24), the consensus levels of the three decision makers are as follows: $GCD(e_1) = 0.9257, GCD(e_2) = 0.904, GCD(e_3) = 0.9777$.

According to Equation (25), the average group consensus level of the three experts is calculated as $GCD\{e_1, e_2, e_3\} = 0.9358$. Since $GCD\{e_1, e_2, e_3\} = 0.9358 < \overline{GCD} = 0.94$, the three experts use the proposed adjustment mechanism to modify their preferences to improve the level of group consensus. Different feedback regulation rules are taken into account to represent the structure according to different preferences:

For $e_1 \in E^H$, $SUG^{(1)} = \left(sug_{ij}^{(1)}\right)_{4\times4}$ represents adjustment advice to experts $e_1$, where $sug_{ij}^{(1)} = \left[\min\left(h_{ij}^{(1)}, h_{ij}^{(1,c)}\right), \max\left(h_{ij}^{(1)}, h_{ij}^{(1,c)}\right)\right]$. The specific $SUG^{(1)} = \left(sug_{ij}^{(1)}\right)_{4\times4}$ is

$$SUG^{(1)} = \begin{bmatrix} [0.5, 0.5] & [0.3, 0.42] & [0.4, 0.51] & [0.6, 0.67] \\ [0.58, 0.7] & [0.5, 0.5] & [0.5, 0.58] & [0.5, 0.74] \\ [0.49, 0.6] & [0.42, 0.5] & [0.5, 0.5] & [0.6, 0.67] \\ [0.4, 0.33] & [0.26, 0.5] & [0.33, 0.4] & [0.5, 0.5] \end{bmatrix}$$

According to $SUG^{(1)} = \left(sug_{ij}^{(1)}\right)_{4\times4}$, experts $e_1$ gives the following adjusted HFPR:

$$H_1^{(1)} = \begin{bmatrix} \{0.5\} & \{0.4\} & \{0.55\} & \{0.65\} \\ \{0.6\} & \{0.5\} & \{0.6\} & \{0.65\} \\ \{0.45\} & \{0.4\} & \{0.5\} & \{0.6, 0.7\} \\ \{0.35\} & \{0.35\} & \{0.4, 0.3\} & \{0.5\} \end{bmatrix}$$

For $e_2 \in E^A$, $SUG^{(2)} = \left(sug_{ij}^{(2)}\right)_{4\times4}$ represents adjustment advice to experts $e_2$, where $sug_{ij}^{(2)} = \left[\min\left(h_{ij}^{(2)}, h_{ij}^{(2,c)}\right), \max\left(h_{ij}^{(2)}, h_{ij}^{(2,c)}\right)\right]$. The specific $SUG^{(2)} = \left(sug_{ij}^{(2)}\right)_{4\times4}$ is

$$SUG^{(2)} = \begin{bmatrix} [1, 1] & \left[\frac{1}{3}, 0.74\right] & [1, 3] & [2, 5] \\ [0.35, 3] & [1, 1] & [1.4, 3] & [2.77, 5] \\ \left[\frac{1}{3}, 1\right] & \left[\frac{1}{3}, 0.71\right] & [1, 1] & [2, 4] \\ [0.2, 0.5] & [0.2, 0.36] & [0.25, 0.5] & [1, 1] \end{bmatrix}$$

According to $SUG^{(2)} = \left(sug_{ij}^{(2)}\right)_{4\times4}$, the adjustment preference relations of the expert $e_2$ is as follows:

$$A_2^{(1)} = \begin{bmatrix} \{1\} & \{1\} & \{2\} & \{2\} \\ \{1\} & \{1\} & \{2\} & \{3\} \\ \left\{\frac{1}{2}\right\} & \left\{\frac{1}{2}\right\} & \{1\} & \{2, 3\} \\ \left\{\frac{1}{2}\right\} & \left\{\frac{1}{3}\right\} & \left\{\frac{1}{2}, \frac{1}{3}\right\} & \{1\} \end{bmatrix}$$

For $e_3 \in E^T$, $SUG^{(3)} = \left( sug_{ij}^{(3)} \right)_{4 \times 4}$ represents adjustment advice to expert $e_3$ with $sug_{ij}^{(3)} = \left[ \min\left( h_{ij}^{(3)}, h_{ij}^{(3,c)} \right), \max\left( h_{ij}^{(3)}, h_{ij}^{(3,c)} \right) \right]$. The specific $SUG^{(3)} = \left( sug_{ij}^{(3)} \right)_{4 \times 4}$ is

$$SUG^{(3)} = \begin{bmatrix} [s_3, s_3] & [(s_2, -0.1), s_3] & [s_3, s_4] & [s_4, (s_5, -0.4)] \\ [s_3, (s_4, 0.1)] & [s_3, s_3] & [(s_4, 0.2), s_5] & [s_5, s_6] \\ [s_2, s_3] & [(s_2, -0.2), s_1] & [s_3, s_3] & [(s_5, -0.5), s_5] \\ [(s_1, 0.4), s_2] & [s_0, s_1] & [s_1, (s_2, -0.5)] & [s_3, s_3] \end{bmatrix}$$

According to $SUG^{(3)} = \left( sug_{ij}^{(3)} \right)_{4 \times 4}$, the adjustment HFLPR of expert $e_3$ is given as follows:

$$T_3^{(1)} = \begin{bmatrix} \{s_3\} & \{s_2\} & \{s_4\} & \{s_5\} \\ \{s_6\} & \{s_3\} & \{s_4\} & \{s_6\} \\ \{s_4\} & \{s_4\} & \{s_3\} & \{s_4, s_5\} \\ \{s_3\} & \{s_2\} & \{s_4, s_3\} & \{s_3\} \end{bmatrix}$$

According to the Equations (16)–(21), the weight vector of the alternative scheme and the consistency degree of the three experts are obtained as follows:

$$w_1^{(1)} = (0.325, 0.425, 0.225, 0.025)^T, \ cl_1^{(1)} = 0.9833$$
$$w_2^{(1)} = (0.375, 0.375, 0.1875, 0.0625)^T, \ cl_2^{(1)} = 0.9896$$
$$w_3^{(1)} = (0.2197, 0.375, 0.2083, 0.125)^T, \ cl_3^{(1)} = 0.9722$$

Furthermore, we obtain group priority weight vector $w_c^{(1)} = (0.3337, 0.3911, 0.2059, 0.07)^T$ by aggregating $w_1^{(1)}, w_2^{(1)}, w_3^{(1)}$ according to IOWA operator; where $\lambda_1^{(1)} = 0.3227$, $\lambda_2^{(1)} = 0.3747, \lambda_3^{(1)} = 0.3026$.

According to Equation (25), the average consensus degree of the group is obtained as $GCD\{e_1, e_2, e_3\} = 0.9401$; where $GCD(e_1) = 0.9399, GCD(e_2) = 0.9514$, and $GCD(e_3) = 0.9289$.

As $GCD\{e_1, e_2, e_3\} = 0.9401 > \overline{GCD} = 0.94$, after a round of adjustment, the HHPRs meet the requirements of group consensus, and then the adjustment stops; therefore, the final group priority weight vector is:

$$w_c^* = w_c^{(1)} = (0.3337, 0.3911, 0.2059, 0.07)^T$$

Thus, the order of alternative rescue schemes is $x_2 \succ x_1 \succ x_3 \succ x_4$. Hence, the rescue scheme $x_2$ is the best choice.

## 5. Discussion

### 5.1. Comparison with Aggregation Operators

The aggregation operator is also a common tool to obtain the decision results in GDM. For the initial priority weight vector $w_1 = (0.15, 0.35, 0.35, 0.15)^T$, $w_2 = (0.3947, 0.3947, 0.1316, 0.0789)^T$ and $w_3 = (0.2643, 0.3191, 0.2643, 0.1524)^T$, we carry out weighted averaging (WA) operator and IOWA operator to aggregate $w_1, w_2, w_3$, and then obtain the group priority weight vector of alternatives, shown in Table 4 and Figure 2. It can be easily found that the ranking results of alternatives are the same, but the priority weight vectors are different for the three methods. The priority weights obtained by the first two methods based on operator aggregations are relatively close, which are quite different with the weight values obtained by the proposed method based on consensus achievement. By implementing consensus reaching process, the decision results are more acceptable to most decision makers. Therefore, the proposed model is more scientific.

**Table 4.** Comparison results for three methods.

|  | **Priority Vectors** | **Ranking Results** |
|---|---|---|
| WA operator | $(0.2697, 0.3546, 0.2486, 0.1271)^T$ | $x_2 \succ x_1 \succ x_3 \succ x_4$ |
| IOWA operator | $(0.2619, 0.3551, 0.255, 0.128)^T$ | $x_2 \succ x_1 \succ x_3 \succ x_4$ |
| Our study | $(0.3337, 0.3911, 0.2059, 0.07)^T$ | $x_2 \succ x_1 \succ x_3 \succ x_4$ |

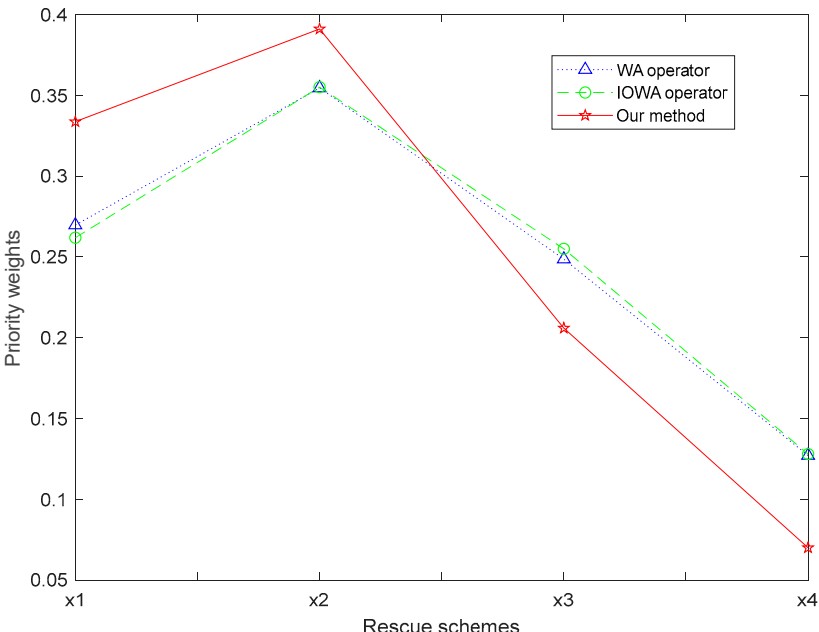

**Figure 2.** The priority weights of four rescue schemes from three methods.

*5.2. Comparison with the Related Studies*

In this section, we mainly discuss comparative analysis with the most relevant studies [34,35]. The specific comparison results are shown in Table 5. It can be seen that the two studies [34,35] can only deal with the GDM problems for numerical type-based HHPRs but cannot deal with the GDM problems for HHPRs including the linguistic-based hesitation preference relations. The proposed model can not only deal with group decision making problems involving HHPRs including linguistic-based hesitate preference relations, but also establish an interactive consensus-building process, making the decision results more acceptable to most decision makers. In the proposed interactive strategy, experts can truly see that the pre-defined consensus level is gradually achieved through their own adjustment, which makes the decision result more acceptable to experts, and therefore more convincing than the decision result obtained with a method [34]. In addition, this paper proposed a method to dynamically adjust the weights of decision experts. According to the information quality of decision experts after adjustment, the new weights of experts can be determined. The dynamic adjustment of decision makers' weights is displayed in Figure 3. From the Figure 3, it can be seen that weights of decision makers have changed over adjustment of preferences. However, the weight values of the decision makers do not change during the decision-making process in the relevant studies [34,35], so it cannot dynamically reflect the quality of preference information provided by the decision makers.

**Table 5.** Comparisons between the proposed method and the existing method.

| Methods | Preference Composition | Consider Dynamically Adjusting Weights of Experts | Consider Individual Consistency | Consider Consensus |
|---|---|---|---|---|
| This study | HFPR + HMPR + HFLPR | Yes | Yes | Yes |
| Zhang and Guo [34] | HFPR + HMPR | No | Yes | No |
| He and Xu [35] | HFPR + HMPR + HPOS | No | Yes | Yes |

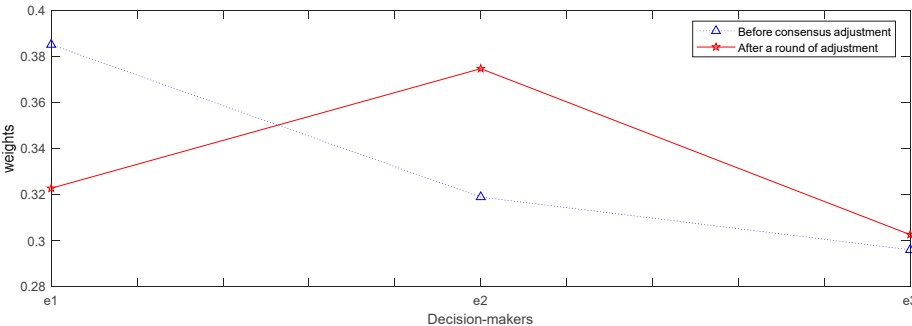

**Figure 3.** The experts' weights are dynamically adjusted during the consensus.

### 5.3. Managerial Implications

The proposed consensus mechanism in group decision making with HHPRs is an effective and practical decision model under uncertain environment. First, the HHPRs (i.e., HMPRs, HFPRs, and HFLPRs) can provide decision makers with more freedom expression of their preference information for the evaluation objects. More specially, the HHPRs used in this paper include not only numeric types but also linguistic type. Thus, the decision maker can choose the preference representation tool suitable for his background knowledge and personality, which shows that the model presented in this paper has better applicability than the existing related models. Then, processing HHPRs with three optimization models is simple and can reduce the loss of information, based on which an interactive consensus-reaching algorithm is built, which can make the decision makers participate in the whole consensus process. This allows for the decision results to be more easily accepted by the decision makers. Furthermore, the consensus model established in this paper is applied to the selection of emergency rescue plan. In this application, the decision makers are in a state of high stress and the rescue scenario is very complex, so it is appropriate for the decision makers to use HHPRs to characterize his preference information. In a word, the model established in this paper can be well applied to the complex decision-making scenarios involving many people. It can be applied not only to daily corporate decisions but also to complex decisions involving multiple people in an organization.

### 5.4. Contributions and Importance of This Study

With the increasing uncertainty of decision-making environments and the complexity of decision-making problems, it is difficult to get scientific decision results only by a single person. In order to solve this problem, group decision making has been widely studied and applied. On the one hand, HHPRs (i.e., HMPRs, HFPRs, and HFLPRs) can be provided to the decision makers to express flexibly preferences for comparison objects. On the other hand, the proposed group decision-making model can deal with HHPRs, including HFLPRs, while existing related research has not been able to address this issue. Therefore, this study further expands the existing research in group decision making with HHPRs.

The purpose of this study is to construct an interactive consensus reaching model with HHPRs in group decision making. By achieving this goal, this study has made some theoretical and practical contributions.

(1) Three optimization models are established to deal with HHPRs, respectively. By the established optimizing models, the normalized weights of the alternatives can easily be obtained. Further, the important weights of decision makers can be obtained according to the obtained consistency deviations of the decision makers.

(2) A consensus reaching model based on the direct consensus framework is developed to guide decision makers to a predetermined level of consensus. In the proposed feedback mechanism-based interaction strategy, decision makers can truly see the whole adjustment process and gradually reach a predefined consensus level through their own adjustment, which makes the decision result more easily accepted and thus more convincing.

(3) A mechanism for dynamically adjusting the important weights of decision makers is established based on the consistency levels of preference information provided by decision makers.

(4) The applicability of the model is illustrated by the study of an emergency case. Through comparative analysis, the performance and advantages of the proposed model are further clarified.

## 6. Conclusions

In this paper, a consensus building mechanism in GDM with HHPRs (i.e., HMPRs, HFPRs, and HFLPRs) is established, which further extends the scope of use of existing studies. Through the consistencies of HHPRs, three optimization models are constructed to obtain the weight vectors of alternatives from HHPRs provided by decision-makers. Then, based on the weight vectors of alternatives, an interactive consensus-building process is established according to direct consensus framework. Through the feedback mechanism, the direction of adjustment is provided to decision-makers. After the repeated interaction process, the decision results with satisfactory consensus level are reached. Finally, the proposed model has been used to deal with a mining rescue scheme selection problem. Through interactive consensus process, the decision makers are involved in the decision-making process and can reach a satisfactory decision result by modifying their personal preferences. Through the study of the coal mine rescue scheme selection, the applicability of the proposed model in complex decision-making environment is further verified. In other words, the model presented in this paper can be used by a company or organization to solve the evaluation and selection problems involving multiple decision makers in a complex environment.

In this article, HHPRs only includes three heterogeneous hesitant preference relations. Thus, a consensus reaching model may be established by considering more heterogeneous preference structures in GDM. Furthermore, the model presented in this paper can also be used in complex decision scenarios such as green supplier selection and choice of venture capital project.

**Funding:** This work was supported by the Doctoral Scientific Research Foundation of Shandong Technology and Business University: No. BS201805.

**Data Availability Statement:** Data is contained within the article.

**Acknowledgments:** I sincerely appreciate the careful work of the editors and thank the reviewers for their suggestions.

**Conflicts of Interest:** The author declares no conflict of interest.

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
