# Peer review of "An Interactive Consensus Model in Group Decision Making with Heterogeneous Hesitant Preference Relations"

_axioms, doi:10.3390/axioms11100517_

Round 1

Reviewer 1 Report

Dear Authors

Initially, I congratulated the authors on the proposed research. The topic is of importance for academic discussion and presents a significant contribution. However, I see opportunities for a few improvements:

1. Although the authors have addressed the contours of the research in the introduction, I believe that the authors could bring the result of more recent research on the topic and/or others that touch on the discussion in the last three years, thus giving a more precise limitation, in this In this sense, I suggest the use of the following works:

The. 1.1 Floriano, C.M., Pereira, V. and Rodrigues, B.e.S. (2022), "3MO-AHP: an inconsistency reduction approach through mono-, multi- or many-objective quality measures", Data Technologies and Applications,. https://doi.org/10.1108/DTA-11-2021-0315

B. 1.2 A Systematic Review of the Applications of Multi-Criteria Decision Aid Methods (1977–2022). Electronics 2022, 11, 1720. https://doi.org/10.3390/electronics11111720

2. I suggest elaborating a little more on the conclusion.

Best Regards

Reviewer

Author Response

Response to Reviewer 1 comments:

General Comments: Initially, I congratulated the authors on the proposed research. The topic is of importance for academic discussion and presents a significant contribution.

Response: Author would like to thank the reviewer for encouraging comments.

Suggestions:

Point 1: Although the authors have addressed the contours of the research in the introduction, I believe that the authors could bring the result of more recent research on the topic and/or others that touch on the discussion in the last three years, thus giving a more precise limitation, in this In this sense, I suggest the use of the following works:

The. 1.1 Floriano, C.M., Pereira, V. and Rodrigues, B.e.S. (2022), "3MO-AHP: an inconsistency reduction approach through mono-, multi- or many-objective quality measures", Data Technologies and Applications,. https://doi.org/10.1108/DTA-11-2021-0315.

  1. 1.2 A Systematic Review of the Applications of Multi-Criteria Decision Aid Methods (1977–2022). Electronics 2022, 11, 1720. https://doi.org/10.3390/electronics11111720.

Response 1: Thanks for your valuable comments and suggestions. These suggestions are very useful for improving the paper. The above two articles have been added as supporting arguments on page 1.

Point 2: I suggest elaborating a little more on the conclusion.

Response 2: Authors would like to thank the reviewer for this constructive comment. Author has now added more contents with conclusions on page 17.

Reviewer 2 Report

The work proposes an interactive consensus reaching model in the group decision making for heterogeneous hesitant preference relations. The paper is interesting, well written and structured. As suggestions for improvement, I highlight the points below:

The Literature Review should be extended, presenting recent studies on group decision making to make satisfactory and reasonable decisions, citing papers such as:

https://ieeexplore.ieee.org/document/9810236

http://dx.doi.org/10.1590/0103-6513.20210011

The Literature Review should highlight trends and publication gaps on the topic. Thus, authors should highlight the main contributions of their article to the academic literature, compared with those papers already published.

The “preliminaries” section is poorly written and should be better explained, as there are several equations without adequate explanations.

The results and conclusions must be improved, explaining the paper's main contributions to the scientific community and society.

Finally, the references are poorly formatted, looking like the numbers have been duplicated.

Author Response

Response to Reviewer 2 comments:

General Comments: The work proposes an interactive consensus reaching model in the group decision making for heterogeneous hesitant preference relations. The paper is interesting, well written and structured.

Response: Authors would like to thank the reviewer for encouraging comments.

Suggestions:

Point 1: The Literature Review should be extended, presenting recent studies on group decision making to make satisfactory and reasonable decisions, citing papers such as:

https://ieeexplore.ieee.org/document/9810236

http://dx.doi.org/10.1590/0103-6513.20210011

The Literature Review should highlight trends and publication gaps on the topic. Thus, authors should highlight the main contributions of their article to the academic literature, compared with those papers already published.

Response 1: Thanks for your valuable comments and suggestions. These suggestions are very useful for improving the paper. Authors have now added the above two references on page 2. Furthermore, the main contributions of this article have been given on pages 2-3.

There are three key contributions of this paper. First, the established consensus reaching process with HHPRs can deal with not only numerical value-based hesitant preference relations (HFPRs and HMPRs) but also linguistic-based hesitant preference relations (HFLPRs), which further expands the application of the HHPRs in the GDM, have some innovative and practical. Second, three optimization models based on consistencies are established to HHPRs. Based on which, an interactive consensus adjustment algorithm is established according to the direct consensus framework. Third, we define the weights of decision makers according to the consistency bias and dynamically adjust the weight of experts based on their information quality in the process of consensus adjustment.

Point 2: The “preliminaries” section is poorly written and should be better explained, as there are several equations without adequate explanations.

Response 2: Authors would like to thank the reviewer for above suggestion for improvement. Authors have now added related explanations for equations on pages 3-5.

Point 3: The results and conclusions must be improved, explaining the paper's main contributions to the scientific community and society.

Response 3: Author would like to thank the reviewer for above suggestion for improvement. On the one hand, author has now added more detailed elaboration of the comparative results in Section 5 (Discussion); on the other hand, the conclusion of this paper has been revised and the contributions of this paper have been emphasized in conclusions on page 17.

Point 4: Finally, the references are poorly formatted, looking like the numbers have been duplicated.

Response 4: Authors would like to thank the reviewer for above suggestion for improvement. Authors have now re-edited the references.

Reviewer 3 Report

In this manuscript, the author  study the group consensus building under HHPRs (i.e., HMPRs,  HFPRs, and HFLPRs). One of the advantages of this method is that it can deal with the  HHPRs including language type, which further extends the scope of use of existing studies. Through the consistencies of HHPRs, three optimization models are constructed to  obtain the weight vectors from HHPRs. Then, based on the direct consensus framework,  an interactive consensus-building process is established.

Through the feedback mechanism, the direction of adjustment is provided to decision-makers. After repeated the  interaction process, the group consensus at a satisfactory level is reached.

Finally, the applicability of the proposed method is verified by a mining rescue scheme selection problem.

In this manuscript ,  the group consensus approach which only includs three heterogeneous hesitant preference relations is considered. Therefore, a consensus approach for GDM with  more heterogeneous preference structures is left for a future research. Moreover,  that consensus reaching method which cabn be established under incomplete HHPRs is also left for a future research.

In this manuscript the author adapted the way of proposing a model in the group decision making for heterogeneous hesitant preference relations and by then he defined the consistencies of three hesitant preference relations. After that he constructed three mathematical programs to obtain the weight vector of the alternatives.  Furthermore, a consensus with satisfactory level was achieved as well as  the practicability and effectiveness of the model are illustrated by using a case study. 

The work seems to be new and the used methods and the model seem to be correct. 

However the English languge need to be checked and the overall style need to better orginazed. 

Author Response

Response to Reviewer 3 comments:

Suggestions:

Point 1: In this manuscript , the group consensus approach which only includs three heterogeneous hesitant preference relations is considered. Therefore, a consensus approach for GDM with more heterogeneous preference structures is left for a future research. Moreover, that consensus reaching method which cabn be established under incomplete HHPRs is also left for a future research.

Response 1: Thanks for your valuable comments and suggestions. These suggestions are very useful for improving the paper. According to the suggestions of reviewer, the last part of the conclusions has been modified for future research directions.

In this article, HHPRs only includes three heterogeneous hesitant preference relations. Thus, future research studies could (1) consider more heterogeneous preference structures in GDM, (2) involve incomplete HHPRs. Besides, some potential applications such as green supplier selection and investment decisions could also be considered.

Point 2: In this manuscript the author adapted the way of proposing a model in the group decision making for heterogeneous hesitant preference relations and by then he defined the consistencies of three hesitant preference relations. After that he constructed three mathematical programs to obtain the weight vector of the alternatives. Furthermore, a consensus with satisfactory level was achieved as well as the practicability and effectiveness of the model are illustrated by using a case study.

Response 2: Authors would like to thank the reviewer for encouraging comments.

Point 3: The work seems to be new and the used methods and the model seem to be correct.

Response 3: Authors would like to thank the reviewer for encouraging comments.

Point 4: However the English languge need to be checked and the overall style need to better orginazed.

Response 4: Authors would like to thank the reviewer for above suggestion for improvement. This paper has now been proofread by a native English.

Reviewer 4 Report

Comments on the paper "An interactive consensus model in group decision making with heterogeneous hesitant preference relations"

Format comments

The formatting of the equations must be corrected.

Content comments

1.       The overall text of the manuscript is quite poor and it needs significant improvement.

2.       The mathematical programming models (eq 11 and 12) are not correct. The optimized function includes variables that are not bounded. An additional explanation also must be added for the construction of the models (eq. 13, 14,16,18). More details should be furnished.  

3.       How the models (eq. 13, 14,16,18) are realized when the decision matrix is constructed applying not only numerical value-based hesitant preference  relations (HFPRs and HMPRs) but also linguistic-based hesitant preference relations (HFLPRs).  How the final consensus is achieved? More details should be furnished.  

4.       What advantages give us the proposed approach comparing to the traditional approaches. More details should be furnished.  

5.       The conclusions must be more closely directed to the novelty aspects of the paper. They are quite trivial in the present version of the manuscript. More details should be furnished.  

Author Response

Response to Reviewer 4 comments:

Format comments: The formatting of the equations must be corrected.

Response: Authors have now read through the paper carefully and corrected all equations.

Suggestions:

Point 1: The overall text of the manuscript is quite poor and it needs significant improvement.

Response 1: Thanks for your valuable comments and suggestions. These suggestions are very useful for improving the paper. The overall text of the manuscript has now been re-edited.

Point 2: The mathematical programming models (eq 11 and 12) are not correct. The optimized function includes variables that are not bounded. An additional explanation also must be added for the construction of the models (eq. 13, 14,16,18). More details should be furnished.

Response 2: Author would like to thank the reviewer for above suggestion for improvement. First, the author has modified Eqs. (11) and (12) on page 6 as follows.

   .              (13)

 .           (14)

Then, the author has now added related additional interpretation for the construction of the models (eq. 13, 14,16,18) on pages 6-8.

Point 3: How the models (eq. 13, 14,16,18) are realized when the decision matrix is constructed applying not only numerical value-based hesitant preference relations (HFPRs and HMPRs) but also linguistic-based hesitant preference relations (HFLPRs). How the final consensus is achieved? More details should be furnished.

Response 3: Authors would like to thank the reviewer for above suggestion for improvement. The Eqs. (13), (14),(16) and (18) are 0-1 mixed goal programming and are realized by MATLAB. Then, based on the obtained individual priority weight vector, we established the consensus reaching process algorithm in Subsection 3.2 on pages9-11.

Point 4: What advantages give us the proposed approach comparing to the traditional approaches. More details should be furnished.

Response 4: Authors would like to thank the reviewer for above suggestion for improvement. Authors have now added advantages of this paper in conclusions on page 17.

Point 5: The conclusions must be more closely directed to the novelty aspects of the paper. They are quite trivial in the present version of the manuscript. More details should be furnished.

Response 5: Thanks for your valuable comments and suggestions. These suggestions are very useful for improving the paper. Author has rewritten the conclusion and added the main contributions of this paper on page 17.

The contributions of this study are as follows:

(1) Three optimization models are established to deal with HHPRs, respectively. By the optimizing models, the normalized weights of the alternatives from HHPRs can easily be obtained. Further, the important weights of decision makers can be obtained according to the obtained consistency deviations of the decision makers.

(2) A direct consensus building process is developed to guide decision makers to a predefined consensus level. This direct process builds a feedback mechanism to address the guidance process. In the proposed interaction strategy, decision makers can truly see the whole adjustment process and gradually reach a predefined consensus level through their own adjustment, which makes the decision result more easily accepted and thus more convincing.

(3) A mechanism for dynamically adjusting the important weights of decision makers is established based on the consistency levels of preference information provided by decision makers.

(4) A contingency case is studied to illustrate the implementation of the proposed approach. The performance and advantages of the proposed method are compared and analyzed.

In this article, HHPRs only includes three heterogeneous hesitant preference relations. Thus, future research studies could (1) consider more heterogeneous preference structures in GDM, (2) involve incomplete HHPRs. Besides, some potential applications such as green supplier selection and investment decisions could also be considered.

Round 2

Reviewer 1 Report

Dear Authors

After reading the authors' responses and checking the implementations proposed by the reviewers, I could see after reading the article that the authors have implemented the suggestions indicated by the reviewers. Therefore, I do not see any further improvements to be made in this version.

I congratulate you on the proposed model.

Best Regards

Author Response

Response to Reviewer 1 Comments

Point 1: After reading the authors' responses and checking the implementations proposed by the reviewers, I could see after reading the article that the authors have implemented the suggestions indicated by the reviewers. Therefore, I do not see any further improvements to be made in this version.

I congratulate you on the proposed model.

Best Regards.

Response 1: Authors would like to thank the reviewer for encouraging comments

Reviewer 2 Report

The authors have endeavored to make all the suggested changes.

In my opinion, the article should be published in the journal.

Author Response

Response to Reviewer 2 Comments

Point 1: The authors have endeavored to make all the suggested changes.

In my opinion, the article should be published in the journal.

Response 1: Authors would like to thank the reviewer for encouraging comments.